# JOINT MULTI-SCALE FORECASTING WITH FFT AND GUMBEL SAMPLING

## ABSTRACT

Multi-scale decomposition has become a mainstream paradigm for time series forecasting. However, existing approaches primarily rely on the input sequence for scale separation, which introduces bias and limits predictive accuracy. In this work, we propose a novel forecasting framework that jointly leverages both input and output sequences to construct a more faithful multi-scale representation. At its core, an FFT-driven adaptive period selection module, augmented with Gumbel sampling, dynamically identifies dominant temporal scales while enabling stochastic yet structured scale exploration during training. To further improve stability and long-horizon robustness, we introduce an adaptive temperature gating mechanism that refines decoder initialization. Extensive experiments on multiple real-world benchmarks demonstrate that our method outperforms state-of-the-art models, providing new insights into temporal decomposition for time series forecasting.

## 1 INTRODUCTION

Time series forecasting is fundamental in numerous applications, including energy management, traffic control, and financial planning(Jin et al., 2024). Accurate prediction of long-horizon sequences remains challenging due to multi-scale temporal dependencies and non-stationarities(Deng et al., 2024; Fan et al., 2024; Tan et al., 2024). Traditional deep learning models often fail to capture these dynamics effectively, limiting their reliability in real-world scenarios(Kim et al., 2025).

Multi-scale decomposition has emerged as a popular approach to model temporal hierarchies, where sequences are separated into components corresponding to different scales(Wang et al., 2024b; Shang et al., 2024). While this paradigm can enhance model expressiveness, existing methods predominantly rely on input-only decomposition(Yu et al., 2024b; Wang et al., 2024c). Such strategies introduce bias, as the derived scales may not accurately reflect future dynamics(Glushkovsky, 2024), and they typically adopt deterministic scale assignments, restricting the exploration of alternative temporal structures(Hu et al., 2024).

These limitations pose two major challenges: first, input-only decomposition may fail to capture scales relevant for long-horizon forecasting(Yang et al., 2024; Jin et al., 2023); second, deterministic or fixed-scale strategies prevent the model from adaptively exploring diverse temporal structures during training, potentially reducing generalization performance across different datasets and prediction horizons(Fan et al., 2024; Zeng et al., 2023).

To address these challenges, we propose a novel forecasting framework that jointly leverages both input and output sequences to construct a more faithful multi-scale representation. At the core of our method is an FFT-driven adaptive period selection module that dynamically identifies dominant temporal scales. We further incorporate a Gumbel sampling mechanism, enabling stochastic yet structured exploration of scales during training. Additionally, an adaptive temperature gating strategy is designed to stabilize the decoder initialization, improving the overall reliability of long-horizon predictions.Our main contributions are summarized as follows:

- We propose a joint multi-scale decomposition framework that integrates both input and output sequences, alleviating the scale bias issue inherent in conventional single-sequence decomposition methods.

- We design an FFT-driven adaptive scale selection module with Gumbel sampling, which enables dynamic and stochastic exploration of dominant temporal scales, enhancing flexibility and robustness in representation learning.

- We introduce a unified network architecture that incorporates an adaptive temperature gating mechanism for decoder initialization, thereby stabilizing the decoding process and improving long-horizon forecasting performance.

## 2 RELATED WORK

### 2.1 TRANSFORMER-BASED MODEL FOR TIME SERIES

Transformer-based architectures remain central to long-term forecasting research. Numerous studies aim to address the inherent limitations of Transformers in temporal sequence modeling. iTransformer (Liu et al., 2023) reformulates the architecture by representing individual series as tokens rather than time steps, thereby capturing inter-sequence dependencies through explicit multivariate correlation modeling. PatchTST (Nie et al., 2023) segments time series into patches and employs channel-independent processing to reduce computational complexity while enabling direct multi-step forecasting. TimesNet (Wu et al., 2023) transforms one-dimensional series into two-dimensional tensors to model both intra-period and inter-period variations adaptively. However, these Transformer variants face a common challenge: while they excel at modeling dependencies between sequences, their autoregressive prediction mechanisms can lead to cumulative errors over long horizons. As demonstrated by DLinear (Zeng et al., 2023), simple linear models sometimes outperform complex Transformers by avoiding sequential error propagation, highlighting the inherent trade-off between the expressive power of attention mechanisms and the robustness of direct prediction strategies.

### 2.2 DECOMPOSITION OF TIME SERIES

Time series decomposition has proven essential for capturing multi-scale temporal patterns. Classical techniques such as seasonal-trend decomposition and wavelet transforms aim to separate signals into distinct components. Recent deep learning approaches have integrated decomposition as explicit modules, with Autoformer (Wu et al., 2021) pioneering auto-correlation mechanisms for periodicity modeling. However, newer methods recognize that fixed decomposition schemes may introduce bias. Leddam (Yu et al., 2024a) introduces learnable decomposition using trainable convolutional kernels that adapt to input-specific patterns, achieving significant error reductions when integrated into existing models. TimeMixer (Wang et al., 2024b) employs multi-scale decomposition with separate mixing operations for seasonal and trend components across different temporal scales. CycleNet (Lin et al., 2024) learns recurrent cycles directly from data to model inherent periodicities. While these learnable approaches represent substantial progress, they still fundamentally depend solely on the input sequence for decomposition, potentially limiting their ability to capture the full complexity of temporal dynamics when historical patterns alone may not sufficiently inform future decomposition structures.

## 3 PROPOSED METHOD

In this section, we present our proposed framework for joint multi-scale time series forecasting. An overview of the framework is illustrated in Figure 1. The framework consists of five key components: the overall architecture, FFT-based decomposition, Gumbel sampling for scale exploration, transformer-based temporal modeling blocks with cross-scale fusion, and adaptive temperature gating.

### 3.1 OVERALL ARCHITECTURE

The proposed model adopts an encoder–decoder structure augmented with a joint multi-scale decomposition module. The encoder embeds temporal patterns at different scales, while the decoder incorporates adaptive initialization and multi-scale context to refine long-horizon predictions.

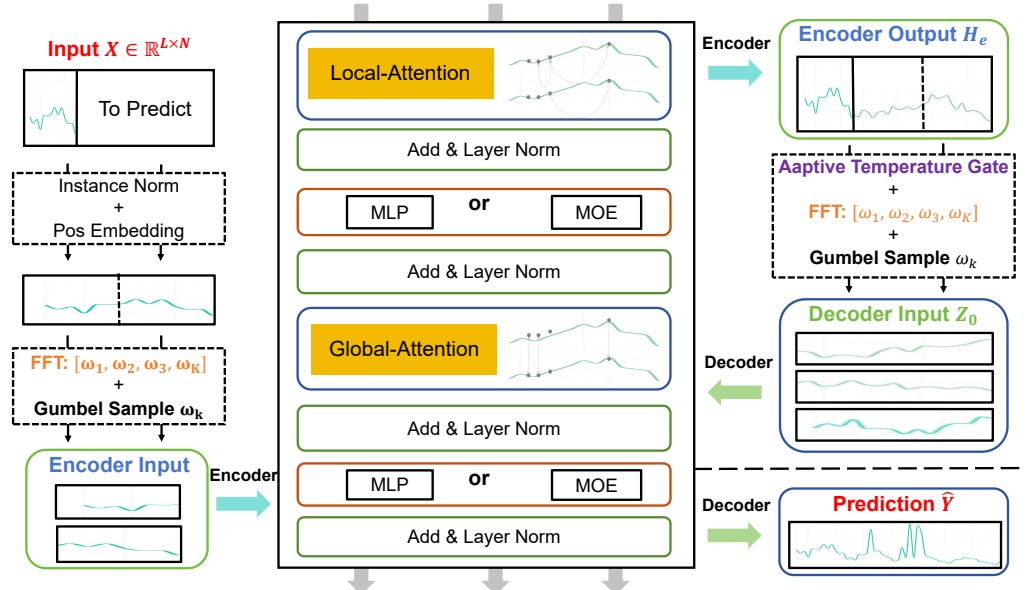

Figure 1: Overall architecture of the proposed framework. The model follows an encoder–decoder design with a joint multi-scale decomposition module.

Let the input sequence be denoted as $\mathbf{X} \in \mathbb{R}^{L \times N}$ and the target horizon as $\mathbf{Y} \in \mathbb{R}^{H \times N}$. The encoder transforms the historical sequence into hidden representations:

$$\mathbf{H}_e = \text{Enc}(\mathbf{X}), \quad \mathbf{H}_e \in \mathbb{R}^{L \times d}. \tag{1}$$

To initialize the decoder, we refine the encoder states using the proposed adaptive temperature gating mechanism, producing the extended initialization matrix:

$$\mathbf{Z}_0 = \mathcal{G}_\tau(\mathbf{H}_e) \in \mathbb{R}^{(L+H) \times d}, \tag{2}$$

where $\mathcal{G}_\tau(\cdot)$ denotes the gating function that adaptively regulates feature sparsity and extends the representation to accommodate both past and future horizons.

The decoder then generates the predictions conditioned on this gated initialization:

$$\hat{\mathbf{Y}} = \text{Dec}(\mathbf{Z}_0). \tag{3}$$

Furthermore, the proposed decomposition and fusion modules are incorporated into both encoder and decoder pathways.

## 3.2 FFT-BASED JOINT MULTI-SCALE DECOMPOSITION

To capture the periodic nature of time series, we employ an FFT-based decomposition module applied at both the encoder and decoder inputs. The encoder processes the input sequence $\mathbf{X}$, where FFT decomposition helps extract multi-scale features. In the decoder, inspired by (Wang et al., 2024a), we combine the generated input and output features and apply FFT to mitigate the auto-correlation issue and enhance representation learning. Formally, we define the frequency-domain transformation as:

$$\mathcal{F}_{\mathbf{I}}(\omega) = \text{FFT}(\mathbf{I}), \quad \mathbf{I} \in \{\mathbf{X}, \mathbf{Z}_0\} \tag{4}$$

The dominant periods are selected as the set of top-$K$ frequency indices:

$$\Omega_{\mathbf{I}} = \text{Topk}_\omega \left( |\mathcal{F}_{\mathbf{I}}(\omega)|, K \right), \tag{5}$$

where $K$ is a hyperparameter. The corresponding temporal periods are then computed as:

$$\mathcal{T}_{\mathbf{I}} = \left\{ \frac{L_{\mathbf{I}}}{\omega} \,\middle|\, \omega \in \Omega_{\mathbf{I}} \right\}, \tag{6}$$

with $L_{\mathbf{X}} = L$ and $L_{\mathbf{Z_o}} = L + H$. These identified periods serve as candidate scale representations for subsequent stochastic exploration using Gumbel sampling, enabling a more faithful multi-scale decomposition of the temporal dynamics.

### 3.3 Gumbel-based Period Selection

We employ a Gumbel-perturbed selection mechanism to dynamically choose dominant periods from the FFT-derived candidates. For each input (encoder or decoder), FFT provides a set of candidate scales $\mathcal{T} = [T_1, \ldots, T_K]$ and corresponding normalized magnitudes $\mathcal{M} = [M_1, \ldots, M_K]$, where

$$M_k = \frac{|\mathcal{F}(\omega_k)|}{\sum_{\omega \in \Omega} |\mathcal{F}_{\mathbf{I}}(\omega)|}. \tag{7}$$

The model also maintains the current period $\tilde{T}$, which is incorporated into the candidate set during each selection step. Before selection, the probabilities are adjusted using a decayed confidence factor $\pi_t$. Specifically, the candidate magnitudes are rescaled as

$$\bar{M}_k = \pi_t \cdot M_k, \quad k = 1, \ldots, K, \tag{8}$$

and the current period is appended with residual probability weight

$$\tilde{M}_{K+1} = 1 - \pi_t. \tag{9}$$

Thus, the final candidate set is

$$\mathcal{T} = [T_1, \ldots, T_K, \tilde{T}], \tag{10}$$

$$\mathcal{M} = [\bar{M}_1, \ldots, \bar{M}_K, \tilde{M}_{K+1}], \tag{11}$$

ensuring that $\sum_{k=1}^{K+1} \tilde{M}_k = 1$. The probability-state $\pi_t$ is decayed at each step:

$$\pi_{t+1} \leftarrow \pi_t \cdot \rho, \tag{12}$$

where $\rho \in (0, 1)$ is a fixed decay hyperparameter. This guarantees that as training proceeds, the selection becomes increasingly stable and biased toward the previously chosen period.

To introduce stochasticity, we generate independent Gumbel noise terms $g_k$ for each candidate and compute perturbed logits:

$$g_k \sim \text{Gumbel}(0, 1), \tag{13}$$

$$\ell_k = \log(\tilde{M}_k + \varepsilon) + g_k, \tag{14}$$

where $\varepsilon > 0$ is a small constant for numerical stability. The selected index is given by the Gumbel-argmax:

$$k^\star = \arg\max_k \ell_k, \tag{15}$$

and the period is updated as

$$\tilde{T} \leftarrow T_{k^\star}. \tag{16}$$

This mechanism is applied separately to both encoder and decoder candidate sets, yielding stateful period selections that evolve across iterations.

Based on the selected periods $\tilde{T}_{\mathbf{I}}$, we construct multi-scale representations by reshaping the input sequence into 3D tensors aligned with the corresponding temporal scales. Formally:

$$\mathbf{H}_{\mathbf{I}} = \text{Reshape}_{\tilde{T}_{\mathbf{I}}}(\text{Pad}(\mathbf{I})), \tag{17}$$

where $\text{Pad}(\cdot)$ denotes zero-padding along the temporal dimension to ensure divisibility by the selected period $\tilde{T}_{\mathbf{I}}$. The reshaped tensor $\mathbf{H}_{\mathbf{I}} \in \mathbb{R}^{d \times \tilde{T}_{\mathbf{I}} \times f_{\mathbf{I}}}$ encodes the input sequence under period $\tilde{T}_{\mathbf{I}}$, with $f_{\mathbf{I}}$ representing the corresponding folding factor. The results $\mathbf{H}_{\mathbf{I}}$ provides multi-period representations that serve as inputs for subsequent temporal modeling.

## 3.4 TRANSFORMER-BASED TEMPORAL MODELING BLOCKS

Building upon the padded multi-period representations $\mathbf{H_I} \in \mathbb{R}^{d \times \tilde{T}_\mathbf{I} \times f_\mathbf{I}}$, each Transformer block sequentially captures intra-period and inter-period dependencies via local attention, a feed-forward (MLP or MoE), a permutation (transpose), global attention, and a second feed-forward. Residual connections and layer normalization are applied around each sublayer.

**Local attention & feed-forward.** Treating $\mathbf{H_I}$ as $f_\mathbf{I}$ non-overlapping segments of length $\tilde{T}_\mathbf{I}$, local self-attention is computed as

$$\mathbf{A}_{\text{local}} = \text{Softmax}\left(\frac{Q_l K_l^\top}{\sqrt{d}}\right) V_l, \qquad Q_l = \mathbf{H_I}W_Q^l, \ K_l = \mathbf{H_I}W_K^l, \ V_l = \mathbf{H_I}W_V^l.$$

The local-refined representation is produced with a residual FFN:

$$\mathbf{H}_{\text{local}} = \mathbf{H_I} + \text{FFN}_l(\mathbf{A}_{\text{local}}).$$

**Adaptive FFN choice.** The position-wise feed-forward network is chosen according to layer depth:

$$\text{FFN}(\cdot) = \begin{cases} \text{MLP}(\cdot), & \text{if layer\_id} < n_{\text{dense}}, \\ \text{MoE}(\cdot), & \text{otherwise}, \end{cases}$$

where the MLP is the standard two-layer gated/activated projection

$$\text{MLP}(x) = \sigma(xW_1 + b_1)W_2 + b_2,$$

and the MoE is a mixture-of-experts with soft gating

$$\text{MoE}(x) = \sum_{k=1}^{K} g_k(x) \cdot \text{Expert}_k(x), \qquad \sum_k g_k(x) = 1.$$

This design uses inexpensive MLPs in early (shallow) blocks for efficiency and higher-capacity MoE in deeper blocks for expressiveness, matching the implementation choice in code.

**Global attention & feed-forward.** We permute (transpose) the local-refined tensor along the period/segment axes:

$$\mathbf{H}_{\text{local}}^\top \in \mathbb{R}^{d \times f_\mathbf{I} \times \tilde{T}_\mathbf{I}}.$$

Global self-attention is applied on the transposed tensor:

$$\mathbf{A}_{\text{global}} = \text{Softmax}\left(\frac{Q_g K_g^\top}{\sqrt{d}}\right) V_g, \quad Q_g = \mathbf{H}_{\text{local}}^\top W_Q^g, \ K_g = \mathbf{H}_{\text{local}}^\top W_K^g, \ V_g = \mathbf{H}_{\text{local}}^\top W_V^g.$$

The global-refined tensor follows with another residual FFN:

$$\mathbf{H}_{\text{global}} = \mathbf{H}_{\text{local}}^\top + \text{FFN}_g(\mathbf{A}_{\text{global}}),$$

where $\text{FFN}_g$ follows the same depth-based MLP/MoE selection rule.

**Final reshape.** After global refinement we transpose back and reshape to the original padded multi-period form:

$$\mathbf{H}_{\text{dual}} = \text{Reshape}\left(\mathbf{H}_{\text{global}}^\top\right) \in \mathbb{R}^{d \times \tilde{T}_\mathbf{I} \times f_\mathbf{I}},$$

and (if needed) any padding is truncated to restore the original unpadded length.

This sequential pipeline (local $\rightarrow$ FFN$_l$ $\rightarrow$ transpose $\rightarrow$ global $\rightarrow$ FFN$_g$ $\rightarrow$ reshape) ensures that intra-period patterns are first captured and enriched before being consolidated into inter-period alignments, while the MLP/MoE scheduling balances computational cost and modeling capacity across depth.

## 3.5 TRANSFORMER-BASED TEMPORAL MODELING BLOCKS

Building upon the padded multi-period representations $\mathbf{H_I} \in \mathbb{R}^{d \times \tilde{T}_\mathbf{I} \times f_\mathbf{I}}$, we design temporal modeling blocks that sequentially capture intra-period and inter-period dependencies. Each block consists of local attention, global attention, and feed-forward layers (MLP or MoE), combined with residual connections and normalization.

**Local Attention and Feed-Forward.** For each period $\tilde{T}_\mathbf{I}$, $\mathbf{H_I}$ is treated as $f_\mathbf{I}$ segments of length $\tilde{T}_\mathbf{I}$. Local self-attention is applied within each segment:

$$\mathbf{A}_{\text{local}} = \text{Softmax}\left(\frac{Q_l K_l^\top}{\sqrt{d}}\right) V_l, \quad Q_l = \mathbf{H_I}W_Q^l, \; K_l = \mathbf{H_I}W_K^l, \; V_l = \mathbf{H_I}W_V^l.$$

The result is refined by a position-wise feed-forward network:

$$\mathbf{H}_{\text{local}} = \mathbf{H_I} + \text{FFN}_l(\mathbf{A}_{\text{local}}),$$

where $\text{FFN}_l$ is an MLP for shallow layers and an MoE for deeper layers.

**Global Attention and Feed-Forward.** To capture inter-period alignment, $\mathbf{H}_{\text{local}}$ is permuted along the period axis:

$$\mathbf{X}_{\text{global}} \in \mathbb{R}^{d \times f_\mathbf{I} \times \tilde{T}_\mathbf{I}},$$

on which global attention is applied:

$$\mathbf{A}_{\text{global}} = \text{Softmax}\left(\frac{Q_g K_g^\top}{\sqrt{d}}\right) V_g, \quad Q_g = \mathbf{X}_{\text{global}}W_Q^g, \; K_g = \mathbf{X}_{\text{global}}W_K^g, \; V_g = \mathbf{X}_{\text{global}}W_V^g.$$

The output is further enhanced via

$$\mathbf{H}_{\text{global}} = \mathbf{X}_{\text{global}} + \text{FFN}_g(\mathbf{A}_{\text{global}}),$$

with $\text{FFN}_g$ chosen adaptively (MLP or MoE).

**Final Fusion.** Finally, $\mathbf{H}_{\text{global}}$ is reshaped back to the original temporal axis:

$$\mathbf{H}_{\text{dual}} = \text{Reshape}(\mathbf{H}_{\text{global}}) \in \mathbb{R}^{d \times \tilde{T}_\mathbf{I} \times f_\mathbf{I}}.$$

This sequential pipeline introduces a hierarchical inductive bias: local modules capture intra-period variations, global modules consolidate inter-period structure, while MLP/MoE feed-forwards adaptively balance efficiency and expressiveness across layers.

## 3.6 ADAPTIVE TEMPERATURE GATING

To further stabilize decoder initialization and control feature sparsity, we design an **Adaptive Temperature Gating** module. Unlike static gating strategies, our method introduces a learnable gating distribution with temperature annealing, which adaptively balances exploration in the early stage and sparsification in later stages.

Formally, given input features $\mathbf{x} \in \mathbb{R}^d$ with learnable gate weights $\mathbf{w} \in \mathbb{R}^d$, the gating temperature at step $t$ is updated as

$$\tau_t = \max\left(\tau_{\min}, \tau_0 \cdot \gamma^t\right), \tag{18}$$

where $\tau_0$ is the initial temperature, $\tau_{\min}$ is the lower bound, and $\gamma \in (0, 1)$ is the decay factor.

**Training.** During training, we adopt the Gumbel-Sigmoid reparameterization to enable differentiable sampling:

$$g_i = \sigma\left(\frac{w_i + \epsilon_i}{\tau_t}\right), \quad \epsilon_i = -\log(-\log(u_i)), \quad u_i \sim \mathcal{U}(0, 1), \tag{19}$$

yielding stochastic gates $\mathbf{g} \in [0, 1]^d$ that regulate the activation of each feature.

**Inference.** At inference, we enforce sparsity by selecting the top-$k$ gates according to a predefined sparsity ratio $s$:

$$k = \lfloor d \cdot (1 - s) \rfloor, \quad \mathcal{I} = \text{TopK}(\sigma(\mathbf{w}), k), \tag{20}$$

and set

$$g_i = \begin{cases} 1, & i \in \mathcal{I}, \\ 0, & \text{otherwise.} \end{cases} \tag{21}$$

**Application.** The final gated output is computed as

$$\mathbf{y} = \mathbf{x} \odot \mathbf{g}, \tag{22}$$

where $\odot$ denotes element-wise multiplication with gates reshaped to match the dimensionality of $\mathbf{x}$.

This adaptive gating strategy regulates the sharpness of feature selection during training and enforces structured sparsity during inference. Integrated into decoder initialization, it mitigates unstable dynamics, improves robustness, and ensures reliable long-horizon forecasting.

## 3.7 Loss Function Design

To ensure multi-scale consistency and mitigate biases introduced by performing FFT solely on the input, we apply frequency-domain supervision on both the encoder and decoder outputs. Let $\mathbf{H_I}$ denote the encoder representation after FFT-based decomposition, and $\mathbf{Z}_0$ the decoder input initialized via the adaptive temperature gating. The dual-level spectral losses are then formulated as

$$\mathcal{L}_{\text{freq-enc}} = \left\| \text{mean}\left(\mathcal{F}_{\mathbf{H_I}}(\omega)_b\right) - \text{mean}\left(\mathcal{F}_{\mathbf{Y}}(\omega)_b\right) \right\|_1, \tag{23}$$

$$\mathcal{L}_{\text{freq-dec}} = \left\| \mathcal{F}_{\mathbf{Z}_0}(\omega)_b - \mathcal{F}_{\mathbf{Y}}(\omega)_b \right\|_1, \tag{24}$$

where $\mathbf{Y}$ is the ground truth sequence and $B$ is the batch size.

In addition, a standard time-domain loss $\mathcal{L}_{\text{time}}$ is applied to decoder predictions. The overall training objective is

$$\mathcal{L} = \mathcal{L}_{\text{freq-enc}} + \mathcal{L}_{\text{freq-dec}} + \alpha \, \mathcal{L}_{\text{time}}, \tag{25}$$

with $\alpha$ controlling the contribution of the time-domain term.

By supervising both encoder and decoder in the frequency domain, our model alleviates multi-scale bias from input-only FFT, preserves periodic structures across scales, and improves long-horizon forecasting. This design extends prior FFT-based loss approaches (Wang et al., 2024a) by enforcing spectral consistency across the full sequence modeling pipeline.

## 4 Experiments

We conduct extensive experiments to evaluate the effectiveness of our proposed framework. Specifically, we aim to answer the following research questions (RQs):

- **RQ1:** Does our method consistently outperform state-of-the-art forecasting models across multiple benchmarks?

- **RQ2:** How do the proposed FFT-based joint decomposition, Gumbel-based period selection, and adaptive temperature gating individually contribute to performance gains?

- **RQ3:** How sensitive is our model's performance to changes in key hyperparameters and architectural choices?

### 4.1 Experimental Setup

**Datasets.** We evaluate on eight widely-used multivariate time series forecasting benchmarks, including **ETT** (Zhou et al., 2021), **ECL**, **Traffic**, **Weather** and **PEMS** (Chen et al., 2001). Each dataset is divided chronologically for training, validation and test.

**Baselines.** We compare against ten strong baselines covering four modeling paradigms: **Transformer-based** (Transformer (Vaswani et al., 2017), Autoformer (Wu et al., 2021), FED-former (Zhou et al., 2022), iTransformer (Liu et al., 2024)); **Convolution-based** (TimesNet (Wu et al., 2023), MICN (Wang et al., 2023)); **Linear** (DLinear (Zeng et al., 2023), TiDE (Das et al., 2023)); and **Frequency-domain** (FreDF (Wang et al., 2024a), FreTS (Yi et al., 2023)).

**Implementation Details.** We implement our framework in PyTorch. The number of candidate periods $K$ is set to 8, and the decay rate $\rho$ for period confidence is 0.9. The initial gating temperature is $\tau_0 = 1.0$ with $\gamma = 0.95$ and $\tau_{\min} = 0.1$. Models are trained with the Adam optimizer (Kingma & Ba, 2014), using a learning rate of $1 \times 10^{-4}$ and batch size 32. We follow standard forecasting horizons $H \in \{96, 192, 336, 720\}$ (and $\{12, 24, 36, 48\}$ for PEMS) and report results using mean squared error (MSE) and mean absolute error (MAE).

Table 1: Comprehensive comparison of forecasting models across multiple datasets. Best results are highlighted in bold. 1st Count shows the number of times each model achieved the best performance.

| Models | | JointMS (Ours) | | FreDF (2025) | | iTransformer (2024) | | FreTS (2023) | | TimesNet (2023) | | MICN (2023) | | TiDE (2023) | | DLinear (2023) | | FEDformer (2022) | | Autoformer (2021) | | Transformer (2017) | |
|---|---|---|---|---|---|---|---|---|---|---|---|---|---|---|---|---|---|---|---|---|---|---|---|
| Metrics | | MSE | MAE | MSE | MAE | MSE | MAE | MSE | MAE | MSE | MAE | MSE | MAE | MSE | MAE | MSE | MAE | MSE | MAE | MSE | MAE | MSE | MAE |
| ETTm1 | 96 | **0.298** | **0.352** | 0.324 | 0.362 | 0.346 | 0.379 | 0.339 | 0.374 | 0.338 | 0.379 | 0.318 | 0.366 | 0.364 | 0.387 | 0.345 | 0.372 | 0.389 | 0.427 | 0.468 | 0.463 | 0.591 | 0.549 |
| | 192 | **0.340** | **0.376** | 0.373 | 0.385 | 0.392 | 0.400 | 0.382 | 0.397 | 0.389 | 0.400 | 0.364 | 0.396 | 0.398 | 0.404 | 0.381 | 0.390 | 0.402 | 0.431 | 0.573 | 0.509 | 0.704 | 0.629 |
| | 336 | **0.372** | **0.395** | 0.402 | 0.404 | 0.427 | 0.422 | 0.421 | 0.426 | 0.429 | 0.428 | 0.398 | 0.428 | 0.428 | 0.425 | 0.414 | 0.414 | 0.438 | 0.451 | 0.596 | 0.527 | 1.171 | 0.861 |
| | 720 | **0.428** | **0.430** | 0.469 | 0.444 | 0.494 | 0.460 | 0.485 | 0.462 | 0.495 | 0.464 | 0.514 | 0.501 | 0.487 | 0.461 | 0.473 | 0.451 | 0.529 | 0.498 | 0.749 | 0.569 | 1.307 | 0.893 |
| ETTm2 | 96 | **0.166** | **0.248** | 0.173 | 0.252 | 0.184 | 0.266 | 0.190 | 0.282 | 0.185 | 0.264 | 0.178 | 0.275 | 0.207 | 0.305 | 0.195 | 0.294 | 0.194 | 0.284 | 0.240 | 0.319 | 0.317 | 0.408 |
| | 192 | **0.234** | **0.290** | 0.241 | 0.298 | 0.257 | 0.315 | 0.260 | 0.329 | 0.254 | 0.307 | 0.240 | 0.317 | 0.290 | 0.364 | 0.283 | 0.359 | 0.264 | 0.324 | 0.300 | 0.349 | 1.069 | 0.758 |
| | 336 | 0.304 | **0.330** | 0.298 | 0.334 | 0.315 | 0.351 | 0.373 | 0.405 | 0.314 | 0.345 | 0.299 | 0.354 | 0.377 | 0.422 | 0.384 | 0.427 | 0.319 | 0.359 | 0.339 | 0.375 | 1.325 | 0.869 |
| | 720 | **0.378** | **0.394** | 0.398 | 0.393 | 0.419 | 0.409 | 0.517 | 0.499 | 0.434 | 0.413 | 0.482 | 0.479 | 0.558 | 0.524 | 0.516 | 0.524 | 0.516 | 0.502 | 0.430 | 0.423 | 2.576 | 1.223 |
| ETTh1 | 96 | **0.372** | **0.395** | 0.382 | 0.400 | 0.390 | 0.410 | 0.399 | 0.412 | 0.422 | 0.433 | 0.383 | 0.418 | 0.479 | 0.464 | 0.396 | 0.410 | 0.377 | 0.418 | 0.423 | 0.441 | 0.796 | 0.691 |
| | 192 | **0.406** | **0.407** | 0.430 | 0.427 | 0.443 | 0.441 | 0.453 | 0.443 | 0.465 | 0.457 | 0.500 | 0.491 | 0.521 | 0.503 | 0.449 | 0.444 | 0.421 | 0.445 | 0.498 | 0.485 | 0.813 | 0.699 |
| | 336 | **0.429** | **0.428** | 0.474 | 0.451 | 0.480 | 0.457 | 0.503 | 0.475 | 0.492 | 0.470 | 0.546 | 0.530 | 0.659 | 0.603 | 0.487 | 0.465 | 0.468 | 0.472 | 0.506 | 0.496 | 1.181 | 0.876 |
| | 720 | **0.451** | **0.443** | 0.463 | 0.462 | 0.484 | 0.479 | 0.596 | 0.565 | 0.532 | 0.502 | 0.671 | 0.620 | 0.893 | 0.736 | 0.516 | 0.513 | 0.500 | 0.493 | 0.477 | 0.487 | 1.182 | 0.885 |
| ETTh2 | 96 | **0.285** | **0.337** | 0.289 | 0.337 | 0.301 | 0.349 | 0.350 | 0.403 | 0.320 | 0.364 | 0.361 | 0.404 | 0.400 | 0.440 | 0.343 | 0.396 | 0.347 | 0.391 | 0.383 | 0.424 | 2.072 | 1.140 |
| | 192 | 0.367 | **0.390** | 0.363 | 0.385 | 0.382 | 0.402 | 0.472 | 0.475 | 0.409 | 0.417 | 0.495 | 0.490 | 0.528 | 0.509 | 0.473 | 0.474 | 0.430 | 0.443 | 0.557 | 0.511 | 5.081 | 1.814 |
| | 336 | **0.374** | **0.424** | 0.419 | 0.426 | 0.430 | 0.434 | 0.564 | 0.528 | 0.449 | 0.451 | 0.671 | 0.588 | 0.643 | 0.571 | 0.603 | 0.546 | 0.469 | 0.475 | 0.470 | 0.481 | 3.564 | 1.475 |
| | 720 | **0.398** | 0.440 | 0.415 | 0.437 | 0.447 | 0.455 | 0.815 | 0.654 | 0.473 | 0.474 | 0.968 | 0.712 | 0.874 | 0.679 | 0.812 | 0.650 | 0.473 | 0.480 | 0.500 | 0.515 | 2.469 | 1.247 |
| ECL | 96 | **0.142** | **0.229** | 0.144 | 0.233 | 0.148 | 0.239 | 0.189 | 0.277 | 0.171 | 0.273 | 0.168 | 0.280 | 0.237 | 0.329 | 0.210 | 0.302 | 0.200 | 0.315 | 0.199 | 0.315 | 0.252 | 0.352 |
| | 192 | **0.155** | **0.255** | 0.159 | 0.247 | 0.167 | 0.258 | 0.193 | 0.282 | 0.188 | 0.289 | 0.177 | 0.289 | 0.236 | 0.330 | 0.210 | 0.305 | 0.207 | 0.322 | 0.215 | 0.327 | 0.266 | 0.364 |
| | 336 | **0.167** | **0.271** | 0.172 | 0.263 | 0.179 | 0.272 | 0.207 | 0.296 | 0.208 | 0.304 | 0.185 | 0.296 | 0.249 | 0.344 | 0.223 | 0.319 | 0.226 | 0.340 | 0.232 | 0.343 | 0.292 | 0.383 |
| | 720 | **0.194** | **0.289** | 0.204 | 0.294 | 0.209 | 0.298 | 0.245 | 0.332 | 0.289 | 0.363 | 0.218 | 0.323 | 0.284 | 0.373 | 0.258 | 0.350 | 0.282 | 0.379 | 0.268 | 0.371 | 0.287 | 0.371 |
| Traffic | 96 | **0.377** | **0.262** | 0.391 | 0.265 | 0.397 | 0.272 | 0.528 | 0.341 | 0.504 | 0.298 | 0.609 | 0.317 | 0.805 | 0.493 | 0.697 | 0.429 | 0.577 | 0.362 | 0.609 | 0.385 | 0.686 | 0.385 |
| | 192 | **0.395** | **0.278** | 0.410 | 0.273 | 0.418 | 0.279 | 0.531 | 0.338 | 0.526 | 0.305 | 0.621 | 0.328 | 0.756 | 0.474 | 0.647 | 0.407 | 0.603 | 0.372 | 0.633 | 0.400 | 0.679 | 0.377 |
| | 336 | **0.403** | **0.288** | 0.424 | 0.280 | 0.432 | 0.286 | 0.551 | 0.345 | 0.540 | 0.310 | 0.641 | 0.342 | 0.762 | 0.477 | 0.653 | 0.410 | 0.615 | 0.378 | 0.647 | 0.398 | 0.663 | 0.361 |
| | 720 | **0.440** | **0.295** | 0.460 | 0.298 | 0.467 | 0.305 | 0.598 | 0.367 | 0.570 | 0.324 | 0.671 | 0.354 | 0.719 | 0.409 | 0.694 | 0.429 | 0.649 | 0.403 | 0.668 | 0.415 | 0.693 | 0.381 |
| Weather | 96 | **0.154** | **0.207** | 0.164 | 0.202 | 0.201 | 0.247 | 0.184 | 0.239 | 0.178 | 0.226 | 0.182 | 0.250 | 0.202 | 0.261 | 0.197 | 0.259 | 0.221 | 0.304 | 0.284 | 0.355 | 0.332 | 0.383 |
| | 192 | **0.195** | **0.249** | 0.220 | 0.253 | 0.250 | 0.283 | 0.223 | 0.275 | 0.227 | 0.266 | 0.234 | 0.301 | 0.242 | 0.298 | 0.236 | 0.294 | 0.275 | 0.345 | 0.313 | 0.371 | 0.634 | 0.539 |
| | 336 | **0.245** | **0.280** | 0.275 | 0.294 | 0.302 | 0.317 | 0.272 | 0.316 | 0.283 | 0.305 | 0.268 | 0.325 | 0.287 | 0.335 | 0.282 | 0.332 | 0.338 | 0.379 | 0.359 | 0.393 | 0.656 | 0.579 |
| | 720 | **0.311** | **0.328** | 0.356 | 0.347 | 0.370 | 0.362 | 0.340 | 0.363 | 0.359 | 0.355 | 0.361 | 0.399 | 0.351 | 0.386 | 0.347 | 0.384 | 0.408 | 0.418 | 0.440 | 0.446 | 0.908 | 0.706 |
| PEMS03 | 12 | 0.075 | 0.180 | **0.068** | **0.172** | 0.069 | 0.175 | 0.083 | 0.194 | 0.082 | 0.188 | 0.087 | 0.203 | 0.117 | 0.225 | 0.122 | 0.245 | 0.123 | 0.248 | 0.239 | 0.365 | 0.107 | 0.209 |
| | 24 | 0.090 | 0.198 | 0.096 | 0.205 | 0.098 | 0.210 | 0.127 | 0.241 | 0.110 | 0.216 | **0.086** | 0.198 | 0.233 | 0.320 | 0.202 | 0.320 | 0.160 | 0.287 | 0.492 | 0.506 | 0.121 | 0.227 |
| | 36 | 0.102 | 0.217 | 0.128 | 0.240 | 0.131 | 0.243 | 0.169 | 0.281 | 0.133 | 0.236 | 0.105 | 0.220 | 0.380 | 0.422 | 0.275 | 0.382 | 0.191 | 0.321 | 0.399 | 0.459 | 0.133 | 0.243 |
| | 48 | 0.150 | 0.265 | 0.161 | 0.269 | 0.164 | 0.275 | 0.204 | 0.311 | 0.146 | 0.251 | **0.120** | 0.235 | 0.536 | 0.511 | 0.335 | 0.429 | 0.223 | 0.350 | 0.875 | 0.723 | 0.144 | 0.253 |
| PEMS08 | 12 | **0.078** | **0.175** | 0.080 | 0.182 | 0.085 | 0.189 | 0.095 | 0.204 | 0.110 | 0.239 | 2.193 | 0.871 | 0.121 | 0.231 | 0.152 | 0.274 | 0.175 | 0.275 | 0.446 | 0.483 | 0.213 | 0.236 |
| | 24 | **0.113** | **0.208** | 0.118 | 0.220 | 0.131 | 0.236 | 0.150 | 0.259 | 0.142 | 0.239 | 0.235 | 0.339 | 0.232 | 0.326 | 0.245 | 0.350 | 0.211 | 0.305 | 0.488 | 0.509 | 0.238 | 0.256 |
| | 36 | **0.145** | **0.232** | 0.161 | 0.258 | 0.182 | 0.282 | 0.202 | 0.305 | 0.167 | 0.258 | 0.197 | 0.300 | 0.379 | 0.428 | 0.344 | 0.417 | 0.250 | 0.338 | 0.352 | 0.513 | 0.263 | 0.277 |
| | 48 | **0.190** | **0.269** | 0.206 | 0.293 | 0.236 | 0.323 | 0.250 | 0.341 | 0.195 | 0.274 | 0.242 | 0.324 | 0.543 | 0.527 | 0.437 | 0.469 | 0.293 | 0.371 | 1.052 | 0.781 | 0.283 | 0.295 |

## 4.2 OVERALL PERFORMANCE

Table 1 summarizes results on nine benchmarks against ten baselines. Improvements are especially strong at long horizons ($H = 720$), where it ranks first on eights datasets with notable improvements (e.g., $8.7\%$ on ETTm1, $8.5\%$ on Weather).

The model remains robust across diverse datasets, consistently outperforming baselines on Traffic and surpassing recent SOTA methods (FreDF, iTransformer, FreTS) in 34/36 settings, with substantial margins over earlier models.

### 4.3 ABLATION STUDY (RQ2)

We conduct ablations to examine the effect of key components: (1) **Multi-scale decomposition (MS)**: removing joint FFT and using input-only; (2) **Period selection (PS)**: replacing Gumbel sampling with deterministic top-$K$; (3) **Confidence decay (CD)**: disabling adaptive temperature gating.

As shown in Table 2, each module contributes to the final performance. Removing multi-scale decomposition yields the largest degradation, underscoring its role in capturing long-term patterns. Period selection and confidence decay provide additional gains in robustness and stability. The full model (JointMS) achieves the lowest errors across all datasets, confirming the effectiveness of our design.

Table 2: Ablation study on four datasets. Results are averaged across forecasting horizons.

| Variant | ETTm1 | | ETTh1 | | Traffic | | PEMS08 | |
|---|---|---|---|---|---|---|---|---|
| | MSE | MAE | MSE | MAE | MSE | MAE | MSE | MAE |
| w/o MS | 0.362 | 0.392 | 0.429 | 0.434 | 0.429 | 0.304 | 0.149 | 0.234 |
| w/o PS | 0.381 | 0.419 | 0.427 | 0.425 | 0.427 | 0.305 | 0.147 | 0.235 |
| w/o CD | 0.373 | 0.402 | 0.430 | 0.422 | 0.410 | 0.292 | 0.150 | 0.242 |
| JointMS | **0.359** | **0.388** | **0.414** | **0.418** | **0.403** | **0.280** | **0.131** | **0.221** |

### 4.4 HYPERPARAMETER SENSITIVITY ANALYSIS

We conduct a sensitivity analysis on two key hyperparameters: the hidden dimension $d_{\text{model}}$ and the period selection parameter $K$.

**Hidden Dimension $d_{\text{model}}$**   We observe that increasing $d_{\text{model}}$ generally improves model performance, as larger hidden dimensions enhance the capacity to capture complex temporal dependencies. However, the improvement exhibits diminishing returns beyond a certain point. Across all datasets, $d_{\text{model}} = 512$ achieves a strong balance between accuracy and computational efficiency, representing the optimal choice for the hidden dimension.

**Period Selection $K$**   We further examine the effect of the number of selected periods $K$. The model performs best with moderate values of $K$, where too small or too large values can degrade performance. Empirically, $K = 4$ consistently yields the lowest errors across datasets, indicating that selecting an appropriate number of dominant periods is crucial for capturing temporal structures effectively.

**Summary**   Overall, the analysis demonstrates that the model is robust to moderate variations in hyperparameters. Choosing $d_{\text{model}} = 512$ and $K = 4$ provides strong performance across diverse datasets, balancing accuracy and efficiency.

## 5 CONCLUSION

In this paper, we introduced a novel forecasting framework that jointly leverages input and output sequences to construct faithful multi-scale representations. Our approach combines FFT-driven adaptive period selection with Gumbel-based stochastic exploration and adaptive temperature gating to capture dominant temporal scales while stabilizing long-horizon predictions. Through extensive experiments on diverse real-world datasets, we demonstrated that our method outperforms existing models in forecasting accuracy and effectively models complex multi-scale temporal dynamics. Our findings highlight the importance of joint decomposition and adaptive scale exploration in improving both the expressiveness and robustness of time series forecasting models.

## REFERENCES

Chao Chen, Karl Petty, Alexander Skabardonis, Pravin Varaiya, and Zhanfeng Jia. Freeway performance measurement system: mining loop detector data. *Transportation research record*, 1748 (1):96–102, 2001.

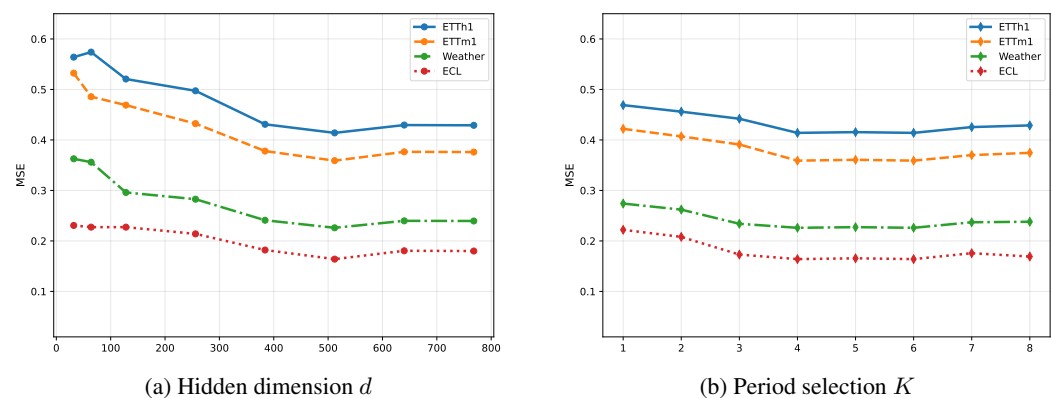

Figure 2: Hyperparameter sensitivity analysis. (a) MSE versus hidden dimension $d$. (b) MSE versus period selection $K$. Intermediate values of $d$ and $K$ achieve the best performance.

Abhimanyu Das, Weihao Kong, Andrew Leach, Shaan Mathur, Rajat Sen, and Rose Yu. Long-term forecasting with tide: Time-series dense encoder. *arXiv preprint arXiv:2304.08424*, 2023.

Jinliang Deng, Feiyang Ye, Du Yin, Xuan Song, Ivor Tsang, and Hui Xiong. Parsimony or capability? decomposition delivers both in long-term time series forecasting. *Advances in Neural Information Processing Systems*, 37:66687–66712, 2024.

Wei Fan, Kun Yi, Hangting Ye, Zhiyuan Ning, Qi Zhang, and Ning An. Deep frequency derivative learning for non-stationary time series forecasting. *arXiv preprint arXiv:2407.00502*, 2024.

Alex Glushkovsky. Dual signal decomposition of stochastic time series. *arXiv preprint arXiv:2508.05915*, 2024.

Yifan Hu, Peiyuan Liu, et al. Adaptive multi-scale decomposition framework for time series forecasting. *arXiv preprint arXiv:2406.03751*, 2024.

Ming Jin, Shiyu Wang, Lintao Ma, Zhixuan Chu, James Y Zhang, Xiaoming Shi, Pin-Yu Chen, Yuxuan Liang, Yuan-Fang Li, Shirui Pan, et al. Time-llm: Time series forecasting by reprogramming large language models. *arXiv preprint arXiv:2310.01728*, 2023.

Ming Jin, Huan Yee Koh, Qingsong Wen, Daniele Zambon, Cesare Alippi, Geoffrey I Webb, Irwin King, and Shirui Pan. A survey on graph neural networks for time series: Forecasting, classification, imputation, and anomaly detection. *IEEE Transactions on Pattern Analysis and Machine Intelligence*, 2024.

Jongseon Kim, Hyungjoon Kim, HyunGi Kim, Dongjun Lee, and Sungroh Yoon. A comprehensive survey of deep learning for time series forecasting: architectural diversity and open challenges. *Artificial Intelligence Review*, 58(7):1–95, 2025.

Diederik P Kingma and Jimmy Ba. Adam: A method for stochastic optimization, 2014.

Shengsheng Lin, Weiwei Lin, Xinyi Hu, Wentai Wu, Ruichao Mo, and Haocheng Zhong. Cyclenet: Enhancing time series forecasting through modeling periodic patterns. *Advances in Neural Information Processing Systems*, 37:106315–106345, 2024.

Yong Liu, Tengge Hu, Haoran Zhang, Haixu Wu, Shiyu Wang, Lintao Ma, and Mingsheng Long. itransformer: Inverted transformers are effective for time series forecasting. *arXiv preprint arXiv:2310.06625*, 2023.

Yong Liu, Tengge Hu, Haoran Zhang, Haixu Wu, Shengzhong Wang, Lintao Ma, and Mingsheng Long. itransformer: Inverted transformers are effective for time series forecasting. In *The Twelfth International Conference on Learning Representations*, 2024.

Yuqi Nie, Nam H Nguyen, Phanwadee Sinthong, and Jayant Kalagnanam. A time series is worth 64 words: Long-term forecasting with transformers. In *International Conference on Learning Representations*, 2023.

Zongjiang Shang, Ling Chen, Binqing Wu, and Dongliang Cui. Ada-mshyper: adaptive multi-scale hypergraph transformer for time series forecasting. *Advances in Neural Information Processing Systems*, 37:33310–33337, 2024.

Mingtian Tan, Mike Merrill, Vinayak Gupta, Tim Althoff, and Tom Hartvigsen. Are language models actually useful for time series forecasting? *Advances in Neural Information Processing Systems*, 37:60162–60191, 2024.

Ashish Vaswani, Noam Shazeer, Niki Parmar, Jakob Uszkoreit, Llion Jones, Aidan N Gomez, Łukasz Kaiser, and Illia Polosukhin. Attention is all you need. *Advances in neural information processing systems*, 30, 2017.

Hao Wang et al. Fredf: Learning to forecast in the frequency domain. *arXiv preprint arXiv:2402.02399*, 2024a.

Huiqiang Wang, Jian Peng, Feihu Huang, Jince Wang, Junyi Chen, and Yifei Xiao. Micn: Multi-scale local and global context modeling for long-term series forecasting. In *The Eleventh International Conference on Learning Representations*, 2023.

Shiyu Wang, Haixu Wu, Xiaoming Shi, Tengge Hu, Huakun Luo, Lintao Ma, James Y Zhang, and Jun Zhou. Timemixer: Decomposable multiscale mixing for time series forecasting. *arXiv preprint arXiv:2405.14616*, 2024b.

Yifan Wang, Peiyuan Liu, et al. Adaptive multi-scale decomposition framework for time series forecasting. *arXiv preprint arXiv:2406.03751*, 2024c.

Haixu Wu, Jiehui Xu, Jianmin Wang, and Mingsheng Long. Autoformer: Decomposition transformers with auto-correlation for long-term series forecasting. *Advances in neural information processing systems*, 34:22419–22430, 2021.

Haixu Wu, Tengge Hu, Yong Liu, Hang Zhou, Jianmin Wang, and Mingsheng Long. Timesnet: Temporal 2d-variation modeling for general time series analysis. In *International Conference on Learning Representations*, 2023.

Runze Yang, Longbing Cao, JIE YANG, et al. Rethinking fourier transform from a basis functions perspective for long-term time series forecasting. *Advances in Neural Information Processing Systems*, 37:8515–8540, 2024.

Kun Yi, Qi Zhang, Wei Fan, Shoujin Wang, Pengyang Wang, Hui He, Ning An, Defu Lian, Longbing Cao, and Zhendong Niu. Frequency-domain mlps are more effective learners in time series forecasting. *Advances in Neural Information Processing Systems*, 36:76656–76679, 2023.

Guoqi Yu, Jing Zou, Xiaowei Hu, Angelica I Aviles-Rivero, Jing Qin, and Shujun Wang. Revitalizing multivariate time series forecasting: Learnable decomposition with inter-series dependencies and intra-series variations modeling. In *International Conference on Machine Learning*, 2024a.

Guoqi Yu, Jing Zou, Xiaowei Hu, Angelica I Aviles-Rivero, Jing Qin, and Shujun Wang. Revitalizing multivariate time series forecasting: Learnable decomposition with inter-series dependencies and intra-series variations modeling. *arXiv preprint arXiv:2402.12694*, 2024b.

Ailing Zeng, Muxi Chen, Lei Zhang, and Qiang Xu. Are transformers effective for time series forecasting? In *AAAI Conference on Artificial Intelligence*, volume 37, pp. 11121–11128, 2023.

Haoyi Zhou, Shanghang Zhang, Jieqi Peng, Shuai Zhang, Jianxin Li, Hui Xiong, and Wancai Zhang. Informer: Beyond efficient transformer for long sequence time-series forecasting. In *Proceedings of the AAAI Conference on Artificial Intelligence*, volume 35, pp. 11106–11115, 2021.

Tian Zhou, Ziqing Ma, Qingsong Wen, Xue Wang, Liang Sun, and Rong Jin. Fedformer: Frequency enhanced decomposed transformer for long-term series forecasting. In *International conference on machine learning*, pp. 27268–27286. PMLR, 2022.

