# OpenReview forum: "Joint Multi-Scale Forecasting with FFT and Gumbel Sampling"
_ICLR.cc/2026/Conference — ICLR 2026 Conference Withdrawn Submission_

### Official Review · Reviewer_4TKe · 2025-10-30

**Soundness:** 2
**Presentation:** 1
**Contribution:** 2
**Rating:** 2
**Confidence:** 4

**Summary:**

The paper proposes a multi-scale time series forecasting model (JointMS), which integrates multi-scale decomposition and an FFT-driven adaptive scale selection module, thereby enhancing the flexibility and robustness of representation learning.

**Strengths:**

S1: The paper proposes a multi-scale time series forecasting model.

S2: The proposed model is evaluated on multiple time series datasets.

**Weaknesses:**

W1 (Motivation): The motivation is unclear. For instance, regarding the first challenge, the authors do not provide a deep analysis of why input-only decomposition fails to capture scales relevant for forecasting. In addition, the paper should explicitly highlight the differences between the proposed approach and existing multi-scale time series modeling methods.

W2 (Writing and Clarity): The overall writing needs improvement, particularly in the methodology section. 1. Some important parts could be better illustrated with figures, e.g., Section 3.3 could include a diagram to clarify the process. 2. Key implementation details are missing—for example, it is not specified which dimensions are modeled by local attention and global attention.

W3 (Formatting Issues): There are several formatting inconsistencies throughout the paper: 1. Some equations include numbering while others do not. 2. The overall paper exceeds the nine-page limit.

W4 (Loss Design): The authors should justify why a frequency-domain loss is introduced and explain why both the encoder and decoder’s frequency representations are constrained using the ground-truth Y.

W5 (Experimental Details): The experimental section lacks important details—for example, the input lengths used for JointMS and the baselines are not specified. The authors should also include comparisons with recent multi-scale methods, such as TimeMixer (ICLR 2024), TimeMixer++ (ICLR 2025), Pathformer (ICLR 2024), and AMD (AAAI 2025).

**Questions:**

See weaknesses.

---

### Official Review · Reviewer_g5Qz · 2025-11-01

**Soundness:** 2
**Presentation:** 2
**Contribution:** 2
**Rating:** 4
**Confidence:** 2

**Summary:**

The authors present an encoder-decoder architecture for multi-scale time series forecasting which combines FFT-driven period selection with Gumbel sampling for temporal scale exploration. The authors evaluate their proposed method against baseline methods and show that it achieves superior performance in prediction accuracy.

**Strengths:**

1. Overall this paper is clearly written and easy to understand.

2. The proposed method using FFT and Gumbel sampling is relatively novel.

3. The authors perform extensive evaluations against baseline methods.

**Weaknesses:**

1. The motivation of Gumbel sampling for scale exploration is not well explained or sufficiently validated.

2. In addition to MAE and MSE, the authors should evaluate their proposed method with MAPE (mean absolute percentage error) which is robust under different scales of the time series values.

3. The authors should also evaluate their proposed method on standard benchmark datasets for time series forecasting, such as the M4 competition dataset.

**Questions:**

1. How does model performance change if we replace Gumbel distribution with other long-tail distributions (e.g., Weibull distribution)?

2. How does model performance change for univariate time series vs. multi-variant time series?

3. How does model performance change if the time series does not contain periodicity?

4. Figure 2 (b) shows that MSE has weak dependence on K. The authors should select K in a logarithmic scale rather than a linear scale between 1 and 8 and select the optimal value of K.

5. Does Gumbel sampling lead to longer time in model training and inference?

---

### Official Review · Reviewer_YvJb · 2025-11-01

**Soundness:** 3
**Presentation:** 2
**Contribution:** 3
**Rating:** 4
**Confidence:** 3

**Summary:**

This paper proposes a joint multi-scale forecasting framework for long-horizon time series prediction. The core idea is a new FFT-based decomposition scheme that operates jointly on both encoder and decoder representations to better capture dominant temporal scales across input and output sequences. The method integrates three components: an FFT-driven adaptive period selection module, a Gumbel sampling mechanism and an adaptive temperature gating strategy.

**Strengths:**

1. The paper introduces a novel use of Gumbel sampling to dynamically select temporal periods, enabling the model to explore multi-scale structures rather than relying on fixed FFT-based choices. This dynamic scale selection helps the model better capture and exploit the underlying periodic structure of time series data
2. The results are generally strong: the method achieves competitive performance across multiple datasets and horizons, and the core components are supported by reasonable ablation results. The experiments give sufficient evidence that the proposed approach is practically effective.
3. The framework is well-engineered and thoroughly specified.

**Weaknesses:**

1. Although the paper describes its approach as a “joint” multi-scale framework, the encoder and decoder apply FFT-based period selection independently, without explicit coupling or scale sharing. The degree of jointness is limited, and the design essentially consists of two parallel decompositions rather than a fully integrated multi-scale structure.
2. The paper applies frequency-domain supervision on both input and output sequences, but the motivation for performing FFT on the output is unclear. While encoder-side FFT captures input patterns, the predicted outputs may exhibit different frequency structures. Supervising output FFT is intended to preserve spectral consistency and mitigate “input-only bias,” yet the paper provides little empirical or intuitive justification for this design.
3. The distinction between the section3.4 and the section3.5 is unclear, and the text repeats similar ideas in both sections, reducing conceptual clarity.
4. A comparison of computational cost against efficient baselines like iTransformer or DLinear is essential to assess practicality and justify the performance gains against the incurred cost.

**Questions:**

1.	Performing FFT on both encoder and output sequences ,specially within the Gumbel sampling for scale exploration,increases computational cost. Could the authors provide runtime or memory analysis?
2.	The paper applies FFT-based supervision on the output sequences, but the motivation for this is not clearly explained. Could the authors clarify why output FFT is necessary beyond encoder-side multi-scale representations, and provide any empirical or intuitive justification for its benefit?

---

### Official Review · Reviewer_zbot · 2025-11-07

**Soundness:** 3
**Presentation:** 3
**Contribution:** 3
**Rating:** 6
**Confidence:** 3

**Summary:**

This paper proposes a novel framework for time series forecasting that addresses the limitations of existing multi-scale decomposition methods, which typically rely solely on input sequences and use deterministic scale assignments. The proposed method, dubbed JointMS (inferred from Table 1), introduces a joint multi-scale decomposition strategy that leverages both input (encoder) and output (decoder) sequences to reduce scale bias.

**Strengths:**

Novel Decomposition Strategy: The idea of using both input and (predicted) output sequences for decomposition is a logical and potentially impactful advancement over input-only methods, which may fail to capture future-relevant scales.

Dynamic Scale Exploration: The integration of FFT with Gumbel sampling for stochastic period selection is a creative way to move beyond rigid, deterministic scale assignments, allowing the model to explore a broader range of temporal dynamics during training.

**Weaknesses:**

Missing Baselines in Main Results: While the related work mentions PatchTST , it is curiously absent from the main performance comparison in Table 1. Given that PatchTST is a leading Transformer-based model, its exclusion is a significant oversight.

Complexity: The proposed architecture is quite complex, involving FFT, Gumbel sampling, adaptive gating, and a specialized Transformer block with both local/global attention and MLP/MoE switches. While effective, it raises questions about computational overhead and ease of implementation compared to simpler models like DLinear.

**Questions:**

Missing SOTA: Why was PatchTST excluded from Table 1? How does JointMS compare to it directly?

Clarification on Loss: Regarding Equation (24), $\mathcal{L}_{freq-dec} = ||\mathcal{F}_{Z_0}(\omega)_b - \mathcal{F}_Y(\omega)_b||_1$. Since $Z_0$ is the input to the decoder (derived from encoder states via gating), why should its frequency spectrum match the ground truth future $Y$ directly? Shouldn't this loss apply to the decoder's output $\hat{Y}$ instead?

Computational Cost: Can you provide metrics on training time and memory usage compared to key baselines like iTransformer and standard Transformers? The added complexity of FFTs and Gumbel sampling at each step might be significant.

---

### Note · Authors · 2025-12-01

I have read and agree with the venue's withdrawal policy on behalf of myself and my co-authors.